# 1 Impact of refreezing melt ponds on Arctic sea ice basal growth

- Daniela Flocco (1), Daniel L. Feltham (1), David Schröeder (1), Michel Tsamados (2)
- (1) Centre for Polar Observations and Modelling Department of Meteorology, University of Reading
- (2) Centre for Polar Observations and Modelling Department of Earth Sciences, University College
   London
- Corresponding author: d.flocco@reading.ac.uk

## 9 1. Abstract 10 Melt ponds forming over the sea ice cover in the Arctic profoundly impact the surface albedo inducing 11 a positive feedback leading to further melting. 12 Here we examine the processes involved in melt pond refreezing and their impact on basal sea ice 13 growth. 14 When ponds freeze, the ice that forms on them insulates the pond trapping it between the sea ice and 15 the ice lid. Trapped melt ponds delay basal sea ice growth in Autumn: ice thickens only after (1) the 16 pond water has been fully frozen and (2) a temperature gradient is established that will conduct heat 17 away from the ocean. Sea ice thickening in the areas where ponds are present is mainly due to the 18 pond's water refreezing. Pan-Arctic simulations with a stand-alone sea ice model and studies with a 19 high-resolution one-dimensional, three-layer refreezing model are used to study the impact on sea ice 20 growth of trapped melt ponds. Basal sea ice growth may be inhibited by up to two months. We estimate 21 an inhibited basal growth of up to 228 km<sup>3</sup>, which represents 25% of the basal sea ice growth estimated 22 by PIOMAS during the months of September and October. The brine not released due to the inhibited 23 basal growth during this period could have implications for the ocean properties and circulation. The 24 impact of trapped melt ponds has not been accounted for so far in any climate model. 25 26 27 28

## **Key points**

| 29 | • | Melt pond refreezing inhibits basal sea ice growth.                                           |
|----|---|-----------------------------------------------------------------------------------------------|
| 30 | • | Internal temperature profile is impacted by the presence of refreezing ponds.                 |
| 31 | • | CICE results show a total over-estimation of basal sea ice growth in Sept - Oct of up to more |
| 32 |   | than 200 km <sup>3</sup> .                                                                    |
|    |   |                                                                                               |

## 34 1. Introduction

- The decline of Arctic Sea ice in the past 25 years has been observed and discussed extensively. In the
- mid-1980s multi-year ice (MYI) accounted for 70% of total winter ice extent, whereas by the end of
- 2012, it had dropped to less than 20% [Stroeve et al., 2014].
- The maintenance of the sea ice system results from a balance of atmosphere ice-ocean thermodynamic and dynamic processes, and the causes of the observed sea ice reduction are complex [Perovich and Richter-Menge, 2009]. Larger areas of open water observed in the summer increase the heat storage in the ocean, leading to increased water temperatures resulting in additional bottom melt of sea ice [Perovich et al., 2008, Perovich et al., 2009, Tsamados et al, 2015] and a consequent delay in winter sea ice formation.
- The solar energy input into the ocean is affected both by the high albedo of sea ice compared to seawater (bare sea ice and snow reflectivity can be up to ~85%; that of water is ~10% [Perovich, 2009]) and the internal absorption of radiation by sea ice. On these grounds it is straightforward to understand the importance of features such as melt ponds that form during spring from snow and ice melt because they lower the total sea ice albedo by up to 20% [Perovich et al. 2002].
- Flocco et al. [2012] performed a number of sensitivity studies to evaluate the impact of including melt ponds in the sea ice component of Global Climate Models (GCMs hereafter) showing a decrease of up to 30% in the surface albedo over the summer months and an average decrease in the September sea ice volume of 40%. Schroeder et al. [2014] showed that the observed September sea ice extent can be skilfully predicted from the modelled spring melt pond fraction in May-June calculated with the model developed by Flocco et al. [2010, 2012].
- Refreezing ponds are difficult to observe because they appear at a time of year when sampling may be challenging. In Figure 1 though, we show one of the few available images of a refreezing pond. This is a snapshot of a video taken in September 2015 during an expedition carried out by Florida University led by David Kadko. The video was taken by William Schmoker, as part of the PolarTREC Program, in the North Canada Basin, north of Barrow, Alaska, by lowering a camera into the ice; it shows a refreezing pond of ~30 cm depth with dendrites of length ~12 cm.

While melt ponds enhance sea ice melting rates over summer, they also inhibit basal sea ice formation 62 during their refreezing. When melt ponds refreeze they have two effects on the internal temperature 63 profile of the ice: latent heat release inhibits cooling of the surrounding ice and salt is released in the 64 trapped pond, lowering its freezing temperature. Even once the pond is completely frozen, basal ice 65 growth cannot start until a negative temperature gradient is established at the ice-ocean interface. This 66 second stage often lasts longer than the pond's refreezing itself. The time that it takes for the 67 temperature gradient to allow ice growth to form depends on the internal temperature profile of the ice 68 and the solid fraction of the ice when the pond has refrozen. Flocco et al. [2015] introduced a high 69 resolution, explicit one-dimensional (1D) model of melt pond refreezing that demonstrates that pond 70 refreezing can delay sea ice basal growth by up to a month in areas where refrozen ponds are present. 71 Current GCMs do not include any explicit treatment of melt pond refreezing.

In this work we show the impact of the presence of refreezing ponds on the internal sea ice temperature profile and assess the impact of melt pond refreezing on the Autumn growth of Arctic sea ice. We do this by combining results from the 1D refreezing model of Flocco et al. [2015] with pan-Arctic simulations from the widely used Los Alamos National Laboratory sea ice model CICE 5.04 [Hunke et al., 2013]. We determine the volume of artificially high basal sea ice growth at the beginning of winter in current GCMs, where the process of pond refreezing is not accounted for.

Section 2 describes the setup of the CICE simulation and our 1D, three-layer refreezing model. Our

results are presented and discussed in section 3, with conclusions presented in section 4.

## 80 2. Methods

We aim to assess the bias introduced in calculations of Autumn basal sea ice volume growth in GCMs 82 caused by the lack of treatment of refrozen ponds, using the combined results from a recently 83 developed a 1D, three-layer model of refreezing melt ponds [Flocco et al., 2015] and the CICE sea ice 84 model.

# 85 *2.1 CICE setup*

CICE is a dynamic-thermodynamic sea ice model designed for inclusion in a global climate model.
Applying the prognostic melt pond model [Flocco et al., 2012] we performed a stand-alone sea-ice
simulation for the pan-Arctic region (~40 km grid resolution) over the period 1979 to September 2013

- using NCEP-DOE-2 Reanalyses data as atmospheric forcing. We implemented a prognostic C-shape
- salinity profile for the pond layer to realistically calculate the freezing of ponds [Flocco et al., 2015].
- Otherwise, the model used is the CICE version 5.04 [Hunke et al. 2013] and the setup is the same as
- was used in Schroeder et al. [2014], using anisotropic rheology, but with 15 instead of 5 ice thickness
- categories. The larger number of categories helps reduce large jumps in the ice thickness distribution
- and distribution of melt ponds. The applied melt pond scheme only affects the surface albedo and the
- freshwater flux into the ocean, otherwise the pond layer is virtual and there is no direct impact on the
- temperature profiles in the ice and snow layers.

## 97 2.2 Melt pond refreezing model

- The 1D, three-layer model simulates a layer of sea ice covered by a freezing trapped melt pond with an
- ice lid on its surface. A schematic of the initial condition of the system is shown in Figure 2.
- The model determines the sea ice internal temperature T in the lid and in the ice underneath the pond
- by solving mushy layers equations in the two ice layers [Flocco et al., 2015]:

$$c_{eff} \frac{\partial I}{\partial t} = \frac{\partial}{\partial z} \left( k_{eff} \frac{\partial I}{\partial z} \right) + \frac{\partial F_{net}(z)}{\partial z},$$
 (1)

where  $F_{net}$  is the net radiative flux,  $c_{eff}$  is the effective volumetric heat capacity defined by

$$c_{\text{eff}} = c_i + \frac{T_L(S_{bulk}) - T_L(0)}{\theta^2} L_{def}$$
(2)

[Bitz and Lipscomb, 1999; Feltham et al., 2006], where  $c_i = 1.883 \times 10^6 \text{ J/(m^3 K)}$  is the specific 106 volumetric heat capacity of sea ice,  $T_L(S_{bulk})$  denotes the liquidus (freezing) temperature of sea ice with 107 salinity  $S_{bulk}$  (e.g.  $T_L(0)=0^{\circ}$ C),  $\theta = T - T_L(0)$ , and  $L = 3.014 \times 10^8 \text{ J m}^{-3}$  is the volumetric latent heat of 108 fusion of pure ice [Bailey et al., 2010]. The effective thermal conductivity of sea ice is given by

$$k_{eff} = k_{bi} - (k_{bi} - k_b) \frac{T_L(S_{bulk}) - T_L(0)}{\theta}$$
, (3)

where  $k_{bi}$  and  $k_b$  are, respectively, the conductivities of bubbly ice and brine, given by

$$k_{bi} = \frac{2k_i + k_a - 2V_a(k_i - k_a)}{2k_i + k_a + 2V_a(k_i - k_a)}k_i$$
(4)

(5)

## 112 and

# 113 $k_b = 0.4184(1.25 + 0.030\theta + 0.00014\theta^2)$

[Schwerdtfeger, 1963], where  $k_i = 1.16 (1.91 - 8.66 \times 10^3 (\theta + 2.97 \times 10^{-5} (\theta^2))$  W (m K)<sup>-1</sup> is the 115 conductivity of pure ice [Sakazume and Seki, 1978],  $k_a = 0.03$  W (m K)<sup>-1</sup> is the conductivity of air 116 [Weeks and Ackley, 1986], and we have assumed a constant  $V_a = 0.025$  as the fractional volume of air 117 in sea ice [Timco and Frederking, 1996].

At the ice lid-air interface the model solves a surface energy balance. A double radiation scheme is applied to the three layers of the model [Taylor and Feltham, 2004] to calculate the radiative fluxes. Continuity of temperature is maintained at the interfaces between the internal layers, which are held at their liquidus temperatures. The temperature at the ice-ocean interface is set to the freezing point of the ocean, which depends on the ocean salinity. The ice growth at the top and at the bottom of the trapped pond is calculated from Stefan conditions at both interfaces.

The salinity in the pond is treated semi-analytically and presents maxima in the solutal boundary layers at both ice-pond interfaces where freezing takes place and leads to salt release [Flocco et al., 2015]. A fraction of the salt contained in the refreezing pond is trapped in the growing sea ice layers depending on the sea ice solid fraction at the interface, therefore, in time, the bulk salinity of the lid and of the sea ice at the bottom of the pond evolves. In particular, during every simulation timestep the ice bulk salinity increases creating a gradient in the lid and in the ice underneath the pond.

We performed simulations over 60 days of the Autumn refreezing period starting with forcing from the 1<sup>st</sup> of September, with varying pond depths (10 to 60 cm) and sea ice thicknesses (0.5 to 2.1 m), using a daily NCEP-DOE-2 climatology for atmospheric forcing [Kanamitsu et al., 2013]. In order to apply representative forcing fields, 2-m air temperature and incoming long-wave and short-wave radiation fluxes are spatially averaged over the area of the Arctic Ocean which is covered by melt ponds. For calculating the mean forcing fields each grid point is weighted with its melt pond fraction, given by its climatological mean on the 1<sup>st</sup> of September from our CICE simulation.

# **3. Results**

#### 138 3.1 Pond statistics from CICE simulation

Figure 3 shows a climatology of total and trapped melt pond fraction in the Arctic based on our 35 year 140 long CICE simulation. In agreement with previous studies [Flocco et al., 2010, 2012; Schroeder et al., 141 2014] melt ponds start to form in the second half of May, reach a maximum fraction in mid July (35% 142 coverage) and are mostly covered by an ice lid by the end of August. In the last decade the maximum 143 pond fraction occurs earlier showing that the melting season starts earlier than in the past. Trapped 144 ponds exist in August and September, covering up to 15% of the sea ice area during August but it is 145 worth noticing that the total and the trapped pond fraction show high inter-annual variability. While the 146 maximum total pond fraction is about 10% higher in the later decade than in the 1980s, the trapped 147 pond fraction has decreased. We believe that this decrease is caused by the shift to a predominantly 148 thinner first year ice cover, which tends to become ponded and melt completely rather than retain pond 149 water at its surface.

The relative percentages of pond depths on the 1st of September, averaged over 1979-2013, occurring 151 over each of the 15 ice thicknesses used in CICE is shown in figure 4. The pond depth distribution 152 covers the whole range from a few millimetres up to more than 1 m. 31% of all ponds are shallow 153 ponds (thinner than 10 cm) and 1.9% of the ponds are deeper than 1 m. The average pond depth is 26 154 cm, a typical value found in field experiments [Polashenski, et al., 2012].

#### 155 3.2 Results from the 1D, three-layer model

We use the 1D model of Flocco et al. [2015] to simulate the refreezing of ponds of variable depth over sea ice thicknesses ranging from 0.55 m to 2.10 m (the mean ice thickness in our CICE ice categories 3, 5, 6 and 8) for a period of 60 days. As an example, if we consider a sea ice thickness of 1.05 m and a melt pond of 0.3 m, the pond freezes in 19 days (stage I), then in the following 24 days a temperature gradient in the sea ice allowing basal ice growth is established (stage II), and in the remaining 17 days of the simulation basal sea ice growth takes place (stage III).

The trapped pond reaches a maximum salinity of 71 psu within the upper solutal boundary layer at the163 lid interface and 55 psu in its interior due to the rejection of salt into the pond.

The minimum melt pond depth that we use as threshold to consider the pond to be refrozen (2 cm), is 165 reached before the pond effectively becomes a brine pocket and starts migrating downwards dissolving 166 the ice at the bottom of the pond (see Flocco et al. [2015] for more details). These results are shown in 167 Figure 5 where it can be observed that in our reference case (ice thickness = 1.05 m and pond depth = 168 30 cm), the ablation of the ice at the bottom of the pond (6 cm) is comparable in magnitude to the basal 169 sea ice growth (7 cm). In Figure 5 we also show the basal ablation at the ice ocean interface, which is a 170 process that occurs at the beginning of each simulation while the pond is still freezing and the pond 171 bottom temperature is above the freezing temperature of the sea water. It is interesting to notice that 172 this process is more important for thinner pond depths: once the whole pond has frozen, the total ice 173 thickness is smaller and the ice growth is faster, therefore the ice basal growth exceeds the basal 174 ablation at the ice-ocean interface. The contrary happens for deeper ponds.

If we consider a layer of unponded ice with a starting ice thickness of 1.05 m, we can compare the 176 corresponding sea ice growth with that of the ponded ice of the same thickness. A layer of sea ice 1.05 177 m thick with no pond cover (the "slab case") would grow by 1.4 cm during stage I, 7.7 cm during stage 178 II, plus a subsequent 11.5 cm during stage III, for a total of 20.6 cm in 60 days. Growth of unponded 179 ice occurs at the base of the ice by freezing sea water with a prescribed salinity of 33 psu, a typical 180 value found in the Arctic. By contrast, ice growth of the ponded ice is mainly due to the pond 181 refreezing. In fact, the total thickness of the ice slab of 1.05 m overlaid by a 30 cm pond has a final 182 thickness of 1.37 m: this figure is the sum of the lid growth, the basal growth and ablation at the bottom 183 of the trapped pond during the pond refreezing.

Figure 6 shows the evolution of temperature profile during refreezing for (initially) ponded and unponded sea ice. Refreezing of the trapped pond prevents cooling of the ice beneath the pond, creating a relatively warm layer insulating the ice beneath the pond. Basal ablation at the ocean interface occurring during pond refreezing can overcome the basal growth after the pond has refrozen. In Figure 6 we see that during the refreezing of a 30 cm pond, 7 cm of ice at the ocean interface melts, and then, once a negative temperature gradient is established, only 6 cm of ocean water is frozen during the remaining days of the experiment run.

By extending our calculations to a range of sea ice thicknesses and melt pond depths, we reach a moreextensive set of results presented in Figure 7. Figure 7a shows the basal ice growth for unponded ice

and for ponded ice from the 1D model when considering a layer of sea ice of thickness 1.05 m covered by ponds of depths varying from 0.1 to 0.6 m. For increasing pond depths most of the estimated inhibited basal growth occurs in the second stage of pond refreezing, revealing the importance of simulating the evolution of the temperature profile in the refrozen pond and sea ice. Figure 7b shows the inhibited basal ice growth for ponded ice for the same experiment during stage I and stage II: this is the amount of basal sea ice growth in a slab of ice during time equivalent to stage I and stage II.

In Figure 7a the difference between the unponded and ponded basal growth is equal to that shown in
Figure 7b plus the decreased sea ice basal growth happening in stage III, when basal growth occur in
both cases. The same holds for figure 7c and 7d.

For simulations with a fixed pond depth of 0.3 m over varying sea ice thickness we see that the inhibiting effect of pond refreezing on basal sea ice growth is a function of the underlying ice thickness (Fig. 7c). The ratio of inhibited growth between Stage I and Stage II depends on the initial ice thickness (Fig. 7d). For sea ice thickness greater than 1.4 m basal growth is not observed within the 60 days of model simulation. This is because it takes longer for the heat to be transferred to the ocean through thicker ice and therefore it takes longer for basal growth to start.

# 208 3.3 Combining results from 1D model and CICE

We implemented solutal boundary layers into the melt pond refreezing scheme of Flocco et al. [2012] 210 in the CICE sea ice model. This allows us to determine the inhibited ice growth during the pond 211 refreezing process (Stage I) for our CICE model. We have compared the inhibited basal sea ice growth 212 of the 1D model corresponding to Stage I with our modified CICE model for a range of initial pond and 213 ice thicknesses and the two models present similar results, suggesting that the modifications to CICE 214 are adequate for Stage I (Figure 8). This figure shows a comparison for a 4 ice thickness categories 215 (0.55 m, 1.05 m, 1.35 m, 2.1m) and pond depths ranging from 30 to 60 cm. Thinner ponds have been 216 omitted from the comparison because their life time is overestimated in CICE due to the redistribution 217 of melt water from thicker to thinner ice categories.

In our comparison we only consider Stage I because CICE does not account for the remaining inhibited
growth during Stage II. To combine the results of the two models, we consider again the results shown
in figure 7.

In order to find the total inhibited growth due to the presence of melt ponds we combine the CICE estimate of the inhibited growth in Stage I with the ratio of inhibited growth in Stage II to Stage I from the 1D model (Figure 7b+d). For example, for a pond depth of 30 cm and sea ice thickness of 1.05 m the ratio between the pond freezing time (Stage I) and the time to reach a sea ice growth temperature gradient (Stage II) is about ~ 1:5. The inhibited growth of 1.4 cm during Stage I from the modified CICE model corresponds to an inhibited growth of 7.7 cm during Stage II from the 1D model.

Integrating the CICE results for the inhibited growth corresponding to Stage I over the whole Arctic, 228 we derive a 1979-2013 average value of 23 +/- 9 km<sup>3</sup> with a maximum value of 43 km<sup>3</sup> in 1983. 229 Applying the ratio between Stage I and Stage II from our 1D model for each pond depth and ice 230 thickness category individually, we calculate the total inhibited ice growth. Figure 9 shows the time 231 series of inhibited ice growth where we observe a decreasing trend of -4.4 km<sup>3</sup>/year with considerable 232 interannual variability. The trend arises because the ice cover has thinned and thus the required 233 temperature profile in the ice for refreezing is more quickly established. The thinner ice is also more 234 likely to melt completely.

Over 1979-2013 the mean inhibited ice growth is 126 +/- 55 km<sup>3</sup> with a maximum value of 228 km<sup>3</sup> in
1983. In Figure 10 we show the magnitude of the local inhibited basal ice growth for the extreme year
1983. In this map the grid cell averaged inhibited ice growth is above 10 cm for large areas in the
Canadian Arctic and can reach values as high as 20 cm.

#### 239 4. Conclusion

The presence of refreezing melt ponds delays basal sea ice growth in Autumn: ice only thickens after 241 (1) the pond water has been fully frozen and (2) a temperature gradient within the ice has been 242 established that will conduct heat from the ocean to the atmosphere. In addition, melt ponds cause basal 243 melt because the pond bottom temperature is above the sea water freezing temperature. These 244 processes have not been accounted for so far in any climate model. A stand-alone simulation with a 245 version of the Los Alamos National Laboratory CICE sea ice model shows an overestimated basal sea 246 ice growth of around 20 km<sup>3</sup> due to not considering the first process (pond refreezing). The impact of 247 the second process cannot be determined from CICE, however, using the pond depth distribution from 248 CICE, simulations with a 1D melt pond refreezing model show that the impact on refreezing of the

- second stage (establishing the temperature gradient) is generally stronger, leading to a total
  overestimation of basal ice growth of about 126 km<sup>3</sup> on average and up to 228 km<sup>3</sup> for individual years
  in Autumn over the Arctic basin. These values range from 12% to 23% of the amount of sea ice growth
  estimated from PIOMAS in the same period (September-October) [Zhang et al., 2003].
- The total ice mass balance is not too strongly affected by neglecting the refreezing process. Our CICE simulations show that pond refreezing contributes 113 km3 of ice, which is similar to the mean overestimation of basal ice growth of 128 km3. However, the source of ice volume increase in Autumn is of relevance to related processes: our results suggest existing estimates of negative buoyancy production at the ice—ocean interface due to salt release during sea ice growth have been overestimated in models.
- There is a decreasing volume of refreezing ponds during the last decade, primarily due to the shift from a permanent to a more seasonal Arctic sea ice cover. The majority of ponds are present on thin ice, which are more likely to disappear at the end of summer. While the overestimation of basal ice growth by ignoring pond refreezing has decreased in the last few decades, the overestimated basal growth is not negligible.

# 265 Acknowledgments

- We would like to acknowledge the Natural Environmental Research Council for supporting this work.
- CICE The model with found the supportive documentation at can be 269 http://oceans11.lanl.gov/drupal/CICE. The NCEP data are available for download at 270 www.esrl.noaa.gov/psd/data/gridded/data.ncep.reanalysis2.html.

| 272 | 5. References                                                                                           |
|-----|---------------------------------------------------------------------------------------------------------|
| 273 | Bailey, E., D. L. Feltham, and P. R. Sammonds (2010), A model for the consolidation of rafted sea ice,  |
| 274 | J. Geophys. Res., 115, C04015, doi:10.1029/2008JC005103.                                                |
| 275 | Bitz, C., and W. Lipscomb (1999), An energy-conserving thermodynamic model of sea ice, J.               |
| 276 | Geophys. Res., 104, 15,669–15,677, doi:10.1029/1999JC900100.                                            |
| 277 | Feltham, D. L., Untersteiner, N., Wettlaufer, J. S. and Worster, M. G. (2006) Sea ice is a mushy layer. |
| 278 | Geophysical Research Letters, 33 (14). L14501. ISSN 0094-8276 doi: 10.1029/2006GL026290.                |
| 279 | Flocco, D., D. L. Feltham, and A. K. Turner (2010), Incorporation of a physically based melt pond       |
| 280 | scheme into the sea ice component of a climate model, J. Geophys. Res., 115, C08012,                    |
| 281 | doi:10.1029/2009JC005568.                                                                               |
| 282 | Flocco, D., D. Schroeder, D. L. Feltham, and E. C. Hunke (2012), Impact of melt ponds on Arctic sea     |
| 283 | ice simulations from 1990 to 2007, J. Geophys. Res., doi:10.1029/2012JC008195.                          |
| 284 | Flocco, D., D. L. Feltham, E. Bailey, and D. Schroeder (2015), The refreezing of melt ponds on Arctic   |
| 285 | sea ice, J. Geophys. Res. Oceans, 120, 647-659).                                                        |
| 286 | Hunke, E. C., W. H. Lipscomb, A. K. Turner, N. Jeffery, and S. Elliott (2013), CICE: the Los Alamos     |
| 287 | Sea Ice Model Documentation and Software User's Manual Version 5.0.                                     |
| 288 | Kanamitsu, M., W. Ebisuzaki, J. Woollen, S-K Yang, J.J. Hnilo, M. Fiorino, and G. L. Potter, NCEP-      |
| 289 | DOE AMIP-II Reanalysis (R-2): 1631-1643, Bulletin of the American Meteorological Society,               |
| 290 | (2002, updated 2013).                                                                                   |
| 291 | Laxon, S., N. Peacock, and D. Smith (2003), High interannual variability of sea ice thickness in the    |
| 292 | Arctic region, Nature, 425 (6961), 947–950.                                                             |
| 293 | Perovich D. K., Richter-Menge J. A., (2009) Loss of Sea Ice in the Arctic, Annual Review of Marine      |
| 294 | Science, Vol. 1: 417 – 441.                                                                             |
| 295 | Perovich, D. K., J. A. Richter-Menge, K. F. Jones, and B. Light (2008), Sunlight, water, and ice:       |
| 296 | Extreme Arctic sea ice melt during the summer of 2007, Geophys. Res. Lett., 35, L11501,                 |
| 297 | doi:10.1029/2008GL034007.                                                                               |