# Peer review of "Impact of refreezing melt ponds on Arctic sea ice basal growth"

_The Cryosphere, 2016_

## Referee Comment (RC1) · Anonymous Referee #1 · 17 Jun 2016

This study has two main findings. First, refreezing melt ponds can cause a 25 % reduction in basal ice growth. And second, this process is important and should be represented in large-scale climate models. Unfortunately, the robustness of both results remains unclear to me. Hence, I am unsure whether this paper can be published in The Cryosphere.

First, regarding the quantitative assessment of inhibited basal ice growth, the realism of the model remains unclear. A comparison with observational data sets would be very helpful in this respect. It also remains unclear whether the model setup is realistic with its mixture of ice thicknesses and melt-pond thicknesses. This makes it very hard to judge if the estimate of 228 km3 inhibited ice growth is realistic to within, say, +- 10 % or rather to within +- 80 %.

[Figure]

Second, I doubt whether the process examined here truly is of relevance for climate models. I certainly appreciate that melt ponds are important in model simulations to better capture the evolution of albedo during summer. However, I doubt that an explicit modeling of the refreezing of melt ponds is of similar importance. Climate models usually transfer the surface-melt water underneath the sea ice, while in the present study it stays on top of the sea ice. Hence, for equal amounts of melt water and sea ice the two systems are energetically very similar at the onset of freezing. For similar incoming fluxes they then should overall remain energetically similar throughout the freezing period (unless outgoing fluxes differ substantially, which is unlikely). Hence, energy conservation requires that the amount of ice formed on top of sea ice in the melt-pond model should be very similar to the amount of ice formed underneath sea ice in standard climate models. This then immediately gives that the amount of inhibited basal growth should roughly be equal to the amount of melt-pond water at the onset of the freezing period. According to lines 254/255, this is also found here.

This to me suggests that neither the overall energy balance nor the overall salt flux is significantly affected by the process examined here. While I'd be very happy to be proven wrong, the present paper does not allow one to judge the robustness of this finding.

As I appreciate that the robustness of the results might be difficult to show, it might be worthwhile to consider a transfer of this manuscript to GMD with its stronger focus on the actual model development.

---

## Referee Comment (RC2) · Anonymous Referee #2 · 21 Jun 2016

General comments: Flocco et al discuss the behavior of melt ponds during refreezing and the impacts the process has on refreezing of sea ice in autumn. The authors find that the pond refreezing process delays the onset of ice bottom growth using a 1D model. They suggest that the process is of significant importance based on upscaled GCM results that show lower basal ice growth during September and October by some 25%. The authors argue the loss of bottom growth may be significant to total ice mass balance and to brine fluxes and make some efforts to connect this effect to large scale changes. The result is a recommendation for inclusion of this process in GCM's.

This reviewer agrees with the first reviewer that these conclusions are problematic and that the paper as written does not to give the reader confidence in the methodology or establish confidence in the overall importance of the process. The first reviewer does an excellent job capturing the key issues. The paper tries hard to emphasize

importance but does not thoroughly address the question of "does this actually matter enough to bother trying to get it modeled in a GCM". The authors argue 'yes' – but as the prior reviewer points out, there are issues with building confidence in methodology. This author also feels both the impact on overall mass balance and salinity fluxes (even if correct as stated) are likely negligible to the outcome of a GCM, in the context of other errors and model variability.

This reviewer also feels the importance of the findings that may be available (i.e. their impact on overall sea ice/climate system such as through brine rejection) is not well supported and that the editorial decisions over-reach the available support. Three impacts of this process are candidates for importance, where importance is defined as having a sufficient impact on the sea ice or climate system that omitting the process would meaningfully impact GCM accuracy. (1) Mass balance of the ice - In terms of overall ice mass balance the process seems likely of negligible importance as ice surface growth (in the pond) makes up for any loss in ice bottom growth (and then some). (2) Salt fluxes/negative buoyancy sourcing - The difference in basal growth magnitude or timing might be significant for salt fluxes to the ocean, though this would need to be more carefully considered than the presentation in this paper allows. (3) Entrainment of salt within the sea ice by trapping at the pond bottom (as discussed heavily in Flocco et al., 2015) – the reviewer has extremely extensive experience on ice and has not observed the entrapped high salinity layers at pond bottoms suggested here or in Flocco et al 2015, and believes they are a fiction of the model. In reality salty melt ponds are well connected to the ocean and brine rejection will be possible during growth.

Here are a list of particular areas in the paper's logic that felt need improvement. L89-90 implementation of a C- shape salinity profile for the pond layer as in Flocco 2015 doesn't tell us what salinity you are initializing the model with – some of the ponds in Flocco 2015 are quite salty. Ponds unconnected to the ocean tend to be very low salinity, typically under 2 PSU and often under 0.5 PSU. Many ponds are connected to

the ocean - some directly through large aperatures others through porosity within the ice by summer's end – either can provide an effective means for brine drainage. As such, the trapped salinity layer which is a component of the model is not often found in nature. Further, ponds having high salinities – which do exist – have such high salinity specifically because of a connection to the underlying ocean. The isolation of high salinity water within a refreezing pocket in a melt pond is simply unrealistic. L 140 – The reviewer notes that this agreement with prior studies is all with studies using the same methods, from the same author and same group. The pond coverage and timing is not compared here to independent observations or models. In particular, as this reviewer has noted in prior evaluation of this group's work, melt pond formation in May north of 70 latitude is very rare in observations, but common in this dataset. This isn't a central component of this work, so little needs to be done. This reviewer is still trying to encourage the group to use caution in building a suite of work on shaky melt pond simulations. L148 – this conjecture could be investigated from the model output readily L 150 – 19 days of what atmospheric conditions? L 162 – reviewer is extremely skeptical of these salinity values being realistic for any significant volume of pond water. L215 Thinner ponds omitted – but these are the majority of the ponds you simulate according to Figure 4. Seems like a very selective comparison. What is the impact of the disagreeing pond behavior on the others? L 230 applying a ratio between Stage 1 and II seems sketchy. L 248-49 doesn't the unponded ice have to go through the process of establishing a temperature gradient at the end of summer too? L254 and 255 superscript km3 L256-8 Perhaps. This would be an important finding, however it is conjectural here and unquantified. Is the potential discrepancy in salt significant to something? Does it meaningfully change the state of the sea ice or climate system to miss it? L263 How do you define 'negligible'. The reviewer feels that the overestimated basal ice growth would be negligible if "its impact on the climate system realism in a GCM is negligible." Nothing presented here convinces the reviewer that this is not still the case. L346 Fig 3 – Over what ice area is this percentage? All Arctic sea ice? L363Fig 5 – simulated for where? Under what conditions/latitude. Fig 6... Trapped

pond stays liquid to -4 C. very salty.

Other Comments: There seems to be some recycling of material from Flocco et al., 2015 JGR. The abstract shares several key conclusion sentences and the findings of this paper are very similar to the conclusion of the 2015 paper. It would be helpful to differentiate this paper from that one more clearly.

The confined duration of the simulations within Sept and October only does the reader a disservice, and makes it challenging to understand the real impact of this process. For example it leaves unanswered key questions like: Does the ice 'catch up' in bottom growth later in the year (fresh ponded ice has higher thermal conductivity and lower specific/latent heat capacity)?

L 10-13 odd line returns - appears this should all be 1 paragraph L41 – increase the heat absorption in the ocean (the fate of the heat and how much is stored is not really addressed in the papers cited) L 40-43 run-on sentence. At least needs some commas. L45 – bare ice will not have reflectivity higher than 70%. Dry snow-covered ice is ∼85%. L48 –more up-to date references e.g. Perovich and Polashenski 2012 show even larger impacts of ponds. L50 comma – hereafter), L53 skillfully sp. L53 May-June, (comma) L237 space after 1983. L261 are →is

---

## Author Comment (AC1) · 5 Aug 2016

**Reviewer 1**

*This study has two main findings. First, refreezing melt ponds can cause a 25 % reduction in basal ice growth. And second, this process is important and should be represented in large-scale climate models. Unfortunately, the robustness of both results remains unclear to me. Hence, I am unsure whether this paper can be published in The Cryosphere.*

*First, regarding the quantitative assessment of inhibited basal ice growth, the realism of the model remains unclear. A comparison with observational data sets would be very helpful in this respect.*

The 1D model is based on widely accepted physical principles of local conservation of heat and salt, is robustly formulated in its components [Feltham et al, 2006] and this combination of phases [Flocco et al, 2015, Bailey et al, 2010] has passed peer review already. Of course, detailed comparison with relevant and well-documented observations would be more than nice, but these do not exist. Measurements have been made on a more *ad hoc* basis showing the interior salinity of the refreezing pond predicted in our model is reasonable. Bailey et al. [2010] has shown in a tank experiment at the Hamburg Ship Model Basin (HSVA) and in cold-room laboratory studies that salinities of between 77 and 110 PSU were reached when freezing a layer of water of salinity of 6 and 33 PSU trapped between two sheets of rafted ice. For further details, see:

Bailey, E., D. L. Feltham, and P. R. Sammonds (2010), A model for the consolidation of rafted sea ice, J. Geophys. Res., 115, C04015, doi:10.1029/2008JC005103.

*It also remains unclear whether the model setup is realistic with its mixture of ice thicknesses and melt-pond thicknesses. This makes it very hard to judge if the estimate of 228 km3 inhibited ice growth is realistic to within, say, +- 10 % or rather to within +- 80 %.*

In what way is it unclear? The primary source of uncertainty in determination of the basin impact of inhibited ice growth comes from uncertainty in knowledge of melt pond fraction, and not from the presented work on melt pond refreezing. Our pond depths and thicknesses are within the observed range and the simulations as complete as is possible without full inclusion of the 1D model into CICE (a task that would take some years, and which our manuscript helps motivate).

*Second, I doubt whether the process examined here truly is of relevance for climate models. I certainly appreciate that melt ponds are important in model simulations to better capture the evolution of albedo during summer. However, I doubt that an explicit modeling of the refreezing of melt ponds is of similar importance. Climate models usually transfer the surface-melt water underneath the sea ice, while in the present study it stays on top of the sea ice. Hence, for equal amounts of melt water and sea ice the two systems are energetically very similar at the onset of freezing. For similar incoming fluxes they then should overall remain energetically similar throughout the freezing period (unless outgoing fluxes differ substantially, which is unlikely). Hence, energy conservation requires that the amount of ice formed on top of sea ice in the melt-pond model should be very similar to the amount of ice formed underneath sea ice in standard climate models. This then immediately gives that the amount of inhibited basal growth should roughly be equal to the amount of melt-pond water at the onset of the freezing period. According to lines 254/255, this is also found here.*

While we broadly agree that a physical description of melt pond refreezing is likely to have less impact on the sea ice mass balance in climate simulations than the impact of melt ponds on the albedo evolution, this is not a reason to ignore pond refreezing. The reviewer's reasoning is approximate and misses many important features, many of which are described in the paper. For example, when a climate model puts its sea ice surface melt below the ice it enters the mixed layer,

where it affects the evolution of the mixed layer and, in particular, its depth which in turn has a leading order impact on its temperature and thence basal ice changes. Secondly, the water artificially being frozen onto the bottom of the ice (rather than on top of the ice) is subject to different boundary conditions, which have a leading order impact on the freezing rate. Thirdly, the reviewer neglects the salt budget: the salt trapped in the refreezing pond does not enter the mixed layer until and unless the ice subsequently melts (at a different time and location). Fourthly, and most crucially as demonstrated in our calculations, it is the time taken for evolution of the temperature profile in the refrozen pond and sea ice below the pond to reach a temperature profile suitable for basal freezing that has the largest impact on basal growth. Moreover it is simply not realistic to magically move surface melt to the bottom of the ice; to do so is a climate modelling trick based on expediency. Such tricks are sometimes a practical necessity in climate runs but every trick thus used decreases the physical fidelity of the model and has the potential to materially damage the simulation accuracy. Our work is a step in the direction of removing over-simplified modelling approaches. The impact of pond refreezing on the local mass balance (which is what the reviewer is referring to) is shown in Figure 7 and is not negligible.

*This to me suggests that neither the overall energy balance nor the overall salt flux is significantly affected by the process examined here. While I'd be very happy to be proven wrong, the present paper does not allow one to judge the robustness of this finding.*

As indicated above, the reviewer's simplified conception of the process misses many important features. Rather than speculate on these things, we wonder why the reviewer does not comment on our actual simulations of the refreezing process. He indicates that processes and impacts are unclear, but does not say why he finds these things unclear. We have included figures demonstrating the impact of the refreezing process on the mass balance but these are not commented upon. If the reviewer feels there are methodological issues with our modeling approach then they ought to be stated.

*As I appreciate that the robustness of the results might be difficult to show, it might be worthwhile to consider a transfer of this manuscript to GMD with its stronger focus on the actual model development.*

We believe that our findings are more relevant to understanding of the physics of processes related to melt pond evolution and that our manuscript falls squarely within the remit of The Cryosphere.

**Reviewer 2**

*This reviewer agrees with the first reviewer that these conclusions are problematic and that the paper as written does not to give the reader confidence in the methodology or establish confidence in the overall importance of the process. The first reviewer does an excellent job capturing the key issues. The paper tries hard to emphasize importance but does not thoroughly address the question of "does this actually matter enough to bother trying to get it modeled in a GCM". The authors argue 'yes' – but as the prior reviewer points out, there are issues with building confidence in methodology. This author also feels both the impact on overall mass balance and salinity fluxes (even if correct as stated) are likely negligible to the outcome of a GCM, in the context of other errors and model variability. This reviewer also feels the importance of the findings that may be available (i.e. their impact on overall sea ice/climate system such as through brine rejection) is not well supported and that the editorial decisions over-reach the available support.*

Yes, we attempt to demonstrate the wider importance of our treatment of refreezing melt ponds, but, regardless of whether this case has been made, we believe we have presented a robust modeling study based on sound and published physical principles. Reviewer 1 has not gone into much detail regarding his criticisms so it is not clear what Reviewer 2 is really agreeing with, e.g. given some of his comments below it seems hard to imagine Reviewer 2 agreeing with Reviewer 1's discussed approach of transporting all surface melt to the ice—ocean interface.

*Three impacts of this process are candidates for importance, where importance is defined as having a sufficient impact on the sea ice or climate system that omitting the process would meaningfully impact GCM accuracy. (1) Mass balance of the ice - In terms of overall ice mass balance the process seems likely of negligible importance as ice surface growth (in the pond) makes up for any loss in ice bottom growth (and then some).*

We have calculated the impact on sea ice mass balance of refreezing in our paper. The impact may not be huge but it is not irrelevant and the reviewer's assertion is incorrect.

*(2) Salt fluxes/negative buoyancy sourcing - The difference in basal growth magnitude or timing might be significant for salt fluxes to the ocean, though this would need to be more carefully considered than the presentation in this paper allows.*

This is an interesting topic to explore, and we mention this in our conclusions. But it is also clearly out of scope of this manuscript.

*(3) Entrainment of salt within the sea ice by trapping at the pond bottom (as discussed heavily in Flocco et al., 2015) – the reviewer has extremely extensive experience on ice and has not observed the entrapped high salinity layers at pond bottoms suggested here or in Flocco et al 2015, and believes they are a fiction of the model. In reality salty melt ponds are well connected to the ocean and brine rejection will be possible during growth.*

While we cannot speak of the reviewer's "extremely extensive" experience, we can point to published results demonstrating that such high salinity entrapped layers have been measured in the field and in laboratory experiments. As already noted in the response to Reviewer 1, Bailey et al. [2010] presented results of a number of experiments carried out on rafted sea ice at the HSVA where salinities of up to 110 PSU were measured. More information is available in Bailey's PhD thesis (Bailey 2011, "The consolidation and strength of rafted sea ice", PhD thesis, UCL) and in the references presented below.

Our calculations take account of brine rejection: the porous-medium Rayleigh number is only exceeded a few times during the pond refreezing allowing only a negligible convection within the ice below the pond ($\sim 10^{-3}$ PSU day$^{-1}$). The convection, however, increases once the pond has refrozen allowing the salty layer to desalinate. This means that unless observations were made in the ice under the pond during the pond refreezing, such high salinities in the ice would not be observed. It is perhaps possible that this may have misled the reviewer in his interpretation of observations.

*Here are a list of particular areas in the paper's logic that felt need improvement.*
*L89- 90 implementation of a C- shape salinity profile for the pond layer as in Flocco 2015 doesn't tell us what salinity you are initializing the model with – some of the ponds in Flocco 2015 are quite salty. Ponds unconnected to the ocean tend to be very low salinity, typically under 2 PSU and often under 0.5 PSU.*

The salinity used in the model's initial condition is of 2 PSU.

*Many ponds are connected to the ocean - some directly through large aperatures others through porosity within the ice by summer's end – either can provide an effective means for brine drainage. As such, the trapped salinity layer which is a component of the model is not often found in nature.*

We agree that is quite common that ponds are melted through their bottom and in this case our model does not apply. It is not true that high salinity ponds are only found in these circumstances (below).

*Further, ponds having high salinities – which do exist – have such high salinity specifically because of a connection to the underlying ocean. The isolation of high salinity water within a refreezing pocket in a melt pond is simply unrealistic.*

Trapped melt ponds have been observed during field experiments and the reviewer's assertion (that isolated regions of high salinity water are unrealistic) is wrong. In addition to the mentioned work of Bailey et al [2010], the following works are all referenced in Flocco et al. [2015]:

Bogorodsky et al. (2006), presented a 1D model of ponds refreezing and observed ponds refreezing during a cruise to the North Pole which took place in July-September 2005 on the R/V "Akademik Fedorov". Their model results show salinities of up to 200PSU.

Marchenco et al. (2009), presented results from field experiment carried out during the expedition RV "Academic Fedorov" in September, 2007. They found profiles of temperature of a lid-pond-ice system with which our initial condition are consistent (figure 6 in the referenced paper). During the campaign increased salinities at the pond bottom were observed. In this case the ice under the pond presented a porosity of 0.5 with a hole linking the pond to the ocean.

Bogorodsky, P.V., and A.V. Marchenko carried out a laboratory study on the thermodynamics of two layers of water surrounding a layer of ice. The investigations were carried out on May 19–26, 2009, in the Cold Laboratory of Svalbard University Center. This work, though concentrated more on the elastic viscous ice deformation due to the increase of the water volume, measured increased salinity in the liquid layer due to freezing.

Bogorodsky, P. V., Marchenko A.V., Pnyushkov A. V., (2006), Thermodynamics of freezing melt ponds, Proceedings of the 18th IAHR International Symposium on Ice.

Bogorodsky, P.V., and A.V. Marchenko (2014), Thermodynamic effects accompanying freezing of two water layers separated by a sea ice sheet.

Marchenko, A.V., P. V. Bogorodsky, V. V. Gorbatsky, A. P. Makshtas, A. V. Pnyushkov (2009), Structure and physico mechanical properties of sea ice in the central arctic studied in the expedition arctic -2007, 20th International Conference on Port and Ocean Engineering under Arctic Conditions.

*L 140 – The reviewer notes that this agreement with prior studies is all with studies using the same methods, from the same author and same group.*

This is not true (see reference above), and, even if it were true, then why would this be an issue? The basic physics of ice growth and conservation of salt is pretty well known and our paper does not introduce any new or untested concept.

*The pond coverage and timing is not compared here to independent observations or models.*

The pond coverage used in this paper is calculated by the CICE model and the validation of our melt pond routine has been previously peer reviewed and published: the validation of our melt pond routine calculating the pond fraction we retrieve from the CICE model was described in Flocco et al. [2010 and 2012].

*In particular, as this reviewer has noted in prior evaluation of this group's work, melt pond formation in May north of 70 latitude is very rare in observations, but common in this dataset. This isn't a central component of this work, so little needs to be done. This reviewer is still trying to encourage the group to use caution in building a suite of work on shaky melt pond simulations.*

These comments will naturally suggest to the Editor that there has been some dialogue between the reviewer and the authors of this study. There has not; we have not been approached by anybody offering any material criticism of our work and we have presented our work widely to many sea ice scientists. We would welcome any criticism based on fact. Melt ponds forming early in GCM-style models has been discussed and noted by us in our earlier papers: such an occurrence is hardly outside the realm of typical climate simulation bias, however, and is irrelevant to this submitted manuscript as noted by the reviewer.

*L148 – this conjecture could be investigated from the model output readily*

Our comment was referring to observed sea ice thickness shift to thinner ice. In any case melt pond cover increases in time (see Schroeder et al. [2014]).

*L 150 – 19 days of what atmospheric conditions?*

As stated in lines 130 and following we have used the NCEP-DOE-2 climatology.

*L 162 – reviewer is extremely skeptical of these salinity values being realistic for any significant volume of pond water.*

See above. These sort of high salinity values are found in tank experiments, field experiments, and in independent modelling work by Bogorodsky et al. [2006].

*L215 Thinner ponds omitted – but these are the majority of the ponds you simulate according to Figure 4. Seems like a very selective comparison. What is the impact of the disagreeing pond behavior on the others?*

As stated in the manuscript there are technical reasons why the thinner ponds cannot be compared between CICE and the 1D model (they last longer in CICE due to the redistribution). While the frequency of occurrence of thin ponds is large, their impact on ice growth is small because they freeze very quickly. The ponds with depths between 30 and 60 cm, which are compared in Figure 8, are the dominant ones for the impact on ice growth and salt flux.

*L 230 applying a ratio between Stage 1 and II seems sketchy.*

No climate model yet contains a model of melt ponds on sea ice that accounts for the local energy and salt balances in the pond and during its refreezing. This is why we adopt the hybrid approach of combining our 1D model, which does treat such balances, with the CICE model. We do not consider this approach to be any more "sketchy" than is absolutely necessary and have taken pains to reduce any necessary errors. We show in Figure 7 that the ratio between the inhibited ice growth during the time the pond is refreezing (Stage I) and the time the temperature gradient is establishing (Stage II) depends on melt pond depth and underlying ice thickness. By calculating the ratio independently for each ice thickness category and pond depth (10 cm intervals) and applying the suitable ratio for each grid point and year, our estimate for Stage II from the pan-Arctic CICE simulation is as accurate as possible. Current model physics does not allow us to derive the inhibited ice growth during Stage II from CICE directly.

*L 248-49 doesn't the unponded ice have to go through the process of establishing a temperature gradient at the end of summer too?*

Yes, and this is taken into account, but the temperature profile is not altered by the latent heat release of refreezing the trapped melt pond and evolves in direct response to the surface energy balance with the atmosphere. This is one of the main results of our paper so rather surprising the reviewer only notes it in our conclusions.

*L256-8 Perhaps. This would be an important finding, however it is conjectural here and unquantified. Is the potential discrepancy in salt significant to something? Does it meaningfully change the state of the sea ice or climate system to miss it?*

Our important finding is neither conjectural nor unquantified. We state in the manuscript that the overestimation of basal ice growth in September and October ranges from 12% to 23% of the total ice growth. Basal ice growth is proportional to the salt flux from the ice into the ocean. For the mass balance of sea ice the overestimation of basal ice growth is approximately compensated by the added ice volume from the frozen ponds, but this is not the case for the salinity budget. At the moment the ponds refreeze, there is no salt flux into the ocean. At a later stage some salt from the mushy layer will be released into the ocean, but the amount is small. Thus, not including the physics of melt pond refreezing into climate models leads to an underestimation of salt flux into the ocean of about 20% for the months September and October. While the precise impact of this error in climate simulations is not known (the refreezing model has to be incorporated into the climate model beforehand), the magnitude of the change is likely to modify the ocean stratification and related ocean processes.

*L263 How do you define 'negligible'. The reviewer feels that the overestimated basal ice growth would be negligible if "its impact on the climate system realism in a GCM is negligible." Nothing presented here convinces the reviewer that this is not still the case.*

PIOMAS estimates a basal sea ice growth of about 1000 km$^3$ during the refreezing period. Our simulations range between 12% and 23% of this value: this is our definition of not negligible.

*L346 Fig 3 – Over what ice area is this percentage? All Arctic sea ice?*

The figure shows a pan-Arctic average.

*L363 Fig 5 – simulated for where? Under what conditions/latitude. Fig 6. . .Trapped pond stays liquid to -4 C. very salty.*

In Bailey et al. [2010] temperatures as low as -7.65 $^o$C have been observed in consolidating rafted ice, corresponding to interstitial liquid of about 110 PSU.

*Other Comments: There seems to be some recycling of material from Flocco et al., 2015 JGR. The abstract shares several key conclusion sentences and the findings of this paper are very similar to the conclusion of the 2015 paper. It would be helpful to differentiate this paper from that one more clearly.*

The paragraph describing the model is almost the same as in Flocco et al. [2015]. We choose to repeat the model description to avoid the reader having to look for another paper: the model structure is the same.

In our previous work we have shown thoroughly the development of the physics of the pond refreezing process and speculated on the impact that this could have at an Arctic wide scale. In this work we have compared our model to GCMs results and evaluated the impact of the refreezing process during the two phases characterized by the absence of basal ice growth. We have assessed the time delay of basal ice growth comparing the results CICE model with and without pond refreezing. Our choice of value of area of refreezing ponds in Flocco et al. [2015] was 20%, while in this work we have only used the CICE model results to evaluate the overestimation of basal growth in September and October.

*The confined duration of the simulations within Sept and October only does the reader a disservice, and makes it challenging to understand the real impact of this process. For example it leaves unanswered key questions like: Does the ice 'catch up' in bottom growth later in the year (fresh ponded ice has higher thermal conductivity and lower specific/latent heat capacity)?*

The only available observations of refreezing ponds extend to October. The CICE simulations are run for 20 years. We believe that the ice does catch up since ice growth is faster in areas where ice is thinner, but the effect of the delay of ice growth and of the stratification changes in the ocean layer are unknown at present. Such an investigation is warranted but not possible without substantially altering the CICE model to include a fuller description of the multi-phase physics of the refreezing process.

*L41 – increase the heat absorption in the ocean (the fate of the heat and how much is stored is not really addressed in the papers cited)*

From Perovich et al. [2008]: Calculations indicate that solar heating of the upper ocean was the primary source of heat for this observed enhanced Beaufort Sea bottom melting. An increase in the open water fraction resulted in a 500% positive anomaly in solar heat input to the upper ocean, triggering an ice – albedo feedback and contributing to the accelerating ice retreat.

*L254 and 255 superscript km3*
*L 10-13 odd line returns - appears this should all be 1 paragraph*
*L 40-43 run-on sentence. At least needs some commas.*
*L45 – bare ice will not have reflectivity higher than 70%. Dry snow-covered ice is   85%.*

*L48 –more up-to date references e.g. Perovich and Polashenski 2012 show even larger impacts of ponds.*
*L50 comma – hereafter),*
*L53 skillfully sp.*
*L53 May-June, (comma)*
*L237 space after 1983.*
*L261 are →is*

All these tiny changes will be made if we are allowed to submit a revised manuscript.

---

## Editor Comment (EC1) · C. Haas (Editor) · 11 Aug 2016

Dear Authors, thank you for your careful replies to the reviewer's comments. Your efforts are very much appreciated. Unfortunately at this point I find the arguments quite convolved, and am not sure that the reviewers can be easily convinced to change their views. I will therefore respect the reviewer's unanimous initial evaluation and unfortunately reject publication of your manuscript at this point. However, I hope that the reviews and your already extensive considerations and replies will help to thoroughly rewrite the manuscript and to either re-submit it to TCD or another journal. I agree that the refreezing of melt ponds above and below the ice and its effects on the sea ice mass balance are interesting topics, regardless of their overall relevance for climate models. In some regard those processes are also comparable to the formation and refreezing of slush where flooding has occurred. There is some past work by Maksym

et al. which could be helpful to revise some aspects of your work. Thank you again for your submission and considerations. Best regards Christian Haas